# Fracture Parameters of Cement Mortar with Different Structural Dimensions Under the Direct Tension Test

**DOI:** 10.3390/ma12111850

**Published:** 2019-06-07

**Authors:** Inkyu Rhee, Jun Seok Lee, Young-Sook Roh

**Affiliations:** 1Department of Civil Engineering, Chonnam National University, Gwangju 61186, Korea; 2Institute of Bio-Housing, Chonnam National University, Gwangju 61186, Korea; jslee2080@jnu.ac.kr; 3School of Architecture, Seoul National University of Science and Technology, Seoul 01811, Korea

**Keywords:** cement mortar, fracture energy, fractured area, size effect, characteristic length

## Abstract

In this paper, we measured the fracture properties of cement mortar—which is composed of sand and has a nearly constant diameter—using a direct tension test. Four double-notched mortar bar specimens with different structural dimensions were assessed. The failure load, load-crack mouth opening displacement, and elongation of the gauge length were measured under direct displacement control. The fractured surfaces were scanned and measured so that we could calculate the tensile strength accurately and determine the fracture energy and characteristic length. The average ratio of total fracture energy (G_F_) to specific fracture energy (G_f_) was 1.94; this was lower than the typical value for concrete, of 2.5. The direct tension test showed that the double-notched mortar specimens had a smaller fracture processing zone after the initiation of tensile cracks, so the tail portion of the softening branch was small. This decreased the G_F_/G_f_ ratio. We verified this result based on a nonlinear fracture mechanics simulation and found that it agreed well with our experimental results. We also investigated the size effects of four different scaled specimens while holding the ratio of structural dimension, d, and notch length, a is constant, so that there was no shape effect. The traditional linear elastic fracture mechanics (LEFM) prediction and Bažant’s size effect law yield a gradient closer to 1/2 in the case of relatively large specimens. In the case of our cement mortar specimens, this prediction was not supported, where the value of the slope was 1/0.727. This was unexpected because LEFM predicts strong size effects. One possible explanation for this result is that the size effects of concrete are most often evaluated using a bending test; also, concrete has a larger maximum aggregate size than mortar. Due to the random heterogeneities in aggregate distribution, higher tail energies may be seen for concrete, leading to differences in the G_F_/G_f_ ratio. At the same time, the peak tensile stress could be affected by the relationship between structural dimensions and aggregate size.

## 1. Introduction

A wide variety of experiments have been carried out to explore the fracture properties of concrete, including brittle crack initiation and propagation [1,2,3]. The size effect on geometrically similar structures can be measured in terms of the nominal strength of structure with the effect of the characteristic structure size. In early stage, the classical Weibull theory was adopted for explaining a statistical size effect caused by randomness of material strength. If the size of a quasi-brittle structures becomes sufficiently large compared to material inhomogeneities, the structure becomes perfectly brittle, and if the size becomes sufficiently small, the structure becomes non-brittle because the fracture process zone (FPZ) extends over the whole cross section of the structure [2]. The basic size effect laws that are power laws in terms of structural dimension. In linear elastic fracture mechanics (LEFM), this exponent was estimated as −1/2 when the geometrically similar structures with geometrically similar cracks or notches were considered [2]. The law of deterministic size effect provided a way of bridging two different powers applicable into two adjacent size range. One is strength limit and the other is LEFM limit. According to crack band theory, the FPZ is influenced by the aggregate content and maximum aggregate size [3,4,5]. The structure size at which this bridging transition occurs represents a characteristic size. Extensive research works have been made for last four decades in order to estimate this transition size effect [2]. Since Walsh [2] introduced the doubly logarithmic plot of nominal strength versus size and observed this transitional zone, Bažant, Kim, and Carpenteri [3,6,7] suggested the modified exponent of that of the LEFM theory. On the other hand, the formation of an FPZ would have a significant influence on the total fracture energy (G_F_) of concrete [5]. 

As a binding material, the fracture behavior of cement mortar may be of interest. It will enable us to determine basic fracture properties by canceling out the inclusion effect. Numerous experimental bending tests have been carried out [1,2,3,5,8,9,10,11,12,13,14,15,16,17,18,19,20,21,22] to assess the fracture energy and fracture toughness of cement mortar and concrete. Cementitious materials exhibit a wide range of properties, from relatively ductile to brittle, depending on the water-to-cement ratio, aggregate-to-cement ratio, curing time, environmental condition (wet or dry condition of specimen), boundary condition of fixture, eccentricity of loading [3,5,10,13,14,15]. When investigating the presence of size effect in the material itself experimentally, it seems preferable to carry out uniaxial tension tests. For uniaxial tension tests, this boundary influence has been extensively investigated for rectangular and cylindrical specimens of different sizes using rotating boundary condition [10]. 

These was an effort to decouple a mix of material behavior and structural effects when fixed boundary condition was applied. This wise experimental setup using rotating boundary condition were also proven by a series of lattice analysis for test specimen [10]. The size effect of concrete and sandstone was reported by van Vliet and van Mier [10,14,15]. The tests were carried out on specimens of six different sizes in a scale range of 1:32. In the smallest specimens, failure stress was governed by the aggregates. With increasing size of the specimen, the role of the statistical strength distribution increased as well. The presence of flaws in the specimens determined the ultimate stress if stress/strain gradients of a structural is not overwhelmed [10,14,15]. The basic hypothesis of the cohesive crack model is that for mode I, an FPZ of finite width can be described by a fictitious line crack that transmits normal stress, where this stress is a function of the separation, w. This hyper-elastic type of softening first shows a very steep descending curve and then, at roughly 0.15~0.33 f_t_^’^, a gentler slope [4,5,9]. The tail of the descending curve is very long. This poses severe problems for the measurement of G_F_, which corresponds to the area under the entire curve [5]. As this measurement is designed for concrete material, this tail of the descending branch differs to that of mortar specimens due to the lack of major inclusion. 

When discussing scaling and size effects, it is important to consider that the fracture energy and material strength imply the existence of a characteristic fracture length as a material property. The expression of the characteristic length, l_ch_, in the context of the FPZ, was introduced by Irwin [3,5]. The ratio of the G_F_ and the G_f_ is approximately 2.5 for concrete, and crack bandwidth, w_f_, can be defined as 14G_f_/f_t_^’^ or 5.6G_F_/f_t_^’^, and w_o_ = (1/7)w_f_ for the intercept of the initial tangent with axis of w [5]. These three fracture parameters, G_F_, G_f_, and f_t_^’^ (or l_ch_, l_1_, and f_t_^’^) are sufficient to generate a cohesive model of concrete material. Similar measurements for evaluating the fracture energy and l_ch_ have been reported by many researchers [23,24,25,26,27,28,29,30,31]. 

In this paper, since the fracture parameter measure of cement mortar mostly have been done under three-point bending test, we attempted to determine these three fracture parameters for cement mortar under direct tension test. To this end, it is also important to measure the fracture surface area when estimating tensile strength, rather than using the overall surface area of the specimen. According to van Vliet et al. [3,15], the size effect on strength cannot be fully understood unless the material composition and the specimen geometry, as well as the presence and magnitude of stress/strain gradients is considered. In our case, the statistical aggregate distribution that causes stress/strain gradients, material composition issue was relatively minimized by using cement mortar specimen (material inhomogeneity only considered aside from the specimen shape, load eccentricity, and rotating boundary condition of fixture devices). This may lead that material size effect could not be overruled by these possible influences. Some experimental studies [32] have attempted to use the fracture surface area to calculate fracture energy more precisely. This is essential to quantify the three fracture parameters discussed above. The purpose of this paper is to devise a basic model of cement mortar fractures that considers size effects and to assess fracture properties using uniaxial tensile tests. To this end, we carried out direct tensile tests of cement mortar bar specimens with different structural sizes. The load-crack mouth opening displacement (CMOD) was recorded, in addition to the elongation of the gauge length, using linear variable differential transformers (LVDTs). We then assessed the size effect, based on the tensile strength and fracture energy, in terms of the final CMOD at failure. To accurately measure the fracture energy of the test specimens, each failure surface was scanned using three-dimensional (3D) scanning equipment. This enabled us to quantify the l_ch_, which can be expressed as the surface energy on the cracked surface with respect to the elastic strain energy on the crack band volume. 

## 2. Uniaxial Tension Test of Size Effects

Each specimen was prepared with a 50% water-to-cement ratio and 53% cement-to-sand ratio. A 12 min mixing process was used: 7 min of dry-mixing, the addition of water over 4 min, and then dynamic compaction for 1 min. The specimens were demolded after 1 day of curing under room temperature and placed in water for 27 days. All specimens were tested using the direct tension setup illustrated in Figure 1. The experiments were performed on a hydraulic testing machine with a capacity of 50 kN. The channels for loading both ends of the specimen are made of hardened steel and have polished surfaces. A double-notched tension bar was used to measure the failure load associated with the CMOD. The purpose of this tension test was to quantify the fracture energy (mode I fracture). We also investigated the size dependency of the scaled specimens (T-10, T-20, T-30, and T-40). The ratio of initial notched length to the maximum dimension of a specimen, a/d (d = w_2_), was set to 0.25, as indicated in Table 1. Four group specimens, with ligament areas of 10, 20, 30, and 40 mm^2^, were cast. Each specimen group contained nine sub-specimens. Two LVDTs and two clip gauges (range: ±2.5 mm) were installed to measure the total displacement of the bar and the CMOD at the notched length. The axial deformation referred to in this paper is the average value of the two LVDTs. The CMOD values on either side of the notched part of the specimen showed significant bias when cracks developed in the notched sections, probably due to the instant rotation of the specimen when the crack developed. Therefore, the CMOD referred to in this paper represents the superior result between the two CMODs measured. The average compressive strengths of specimens T-10, T-20, T-30, and T-40 were 66.6, 55.3, 52.4, and 56.7 MPa, respectively, after 28 days of curing. Each specimen was in the form of a 50-mm cube. Failure stress-CMOD plots are depicted in Figure 2 for the T-series specimens. The stresses were calculated based on the initial test area, given by A_o_ = w_2_·b. 

## 3. Uniaxial Cauchy Stress According to 3D Image Analysis

As the fractured area, A_n_, differs from the initial test area, A_o_, for each fractured specimen, 3D image analyses were carried out to measure the actual A_n_. A light-emitting diode (LED)-type remote 3D scanner (smartSCAN; AICON, 2016, Breuckmann, Germany) based on the miniaturized projection technique was used with three different projection angles. The A_n_ of each T-series specimen was measured on a 5.0-M resolution digital image with a minimum precision of 7 μm. The actual fractured surfaces were larger than the initial section, so the fracture stresses were calculated according to the Cauchy stress principle, as indicated in Figure 3. It is important to quantify the mode I fracture energy based on experimental test data. Smaller specimens show relatively larger differences between fractured and initial sections, as indicated in Figure 3e,f.

## 4. Size Effect of Mortar Specimens under Uniaxial Tension

The fracture stress-CMOD graph obtained based on the initial section in Figure 2 was replotted in Figure 4 for a fractured section. The peak values of the fractured surfaces were adjusted to yield lower values. The gauge lengths of specimens T-10, T-20, T-30, and T-40 were 35, 70, 105, and 140 mm, respectively. The tensile strength-CMOD relationships, based on the (a) clip gages and (b) LVDTs with gauge length L_g_, are plotted in Figure 4. The results were similar, indicating that the FPZ was located nearby to the notched zone. The peak strain ranged from 4.18 × 10^−5^ to 8.81 × 10^−5^. The failure strain finally saturated at approximately 1.25 × 10^−3^. The CMODs at failure were 0.0333, 0.091, 0.145, and 0.245 mm for specimens T-10, T-20, T-30, and T-40, respectively.

The experimental results for the peak Piolar-Kirchhoff (PK) stresses and Cauchy stresses, for different structural dimensions, d (from 10–40 mm), are shown in Figure 5a,b. To measure the size effect [4,5,9], we linearized Equation (1) so that the projected reference strength, $f_{t}^{*}=Bf_{t}^{\prime }$ could be evaluated by intersecting the $\sigma_{N}{}^{-2}$-axis in the form of $f_{t}^{*}=1/\sqrt{C}$. Furthermore, the characteristic measure of specimen size, $\lambda_{o}$ can be calculated as C/A. The results of this linear fitting of the PK and Cauchy stress data are plotted in Figure 5c,d, where $Bf_{t}^{\prime }$ = 2.24 Mpa and $\lambda_{o}$ = 7.55 mm for PK stress; $Bf_{t}^{\prime }$ = 3.16 Mpa and $\lambda_{o}$ = 5.24 mm were retrieved. The minimum stress value for each group of specimens was used in the calculations of these parameters based on experimental data. The maximum and average stress data tend to have a negative C value, which is not rational in the case of tensile stress.
(1)$$\sigma_{N}{}^{-2}=\frac{1}{{\left(Bf_{t}^{\prime }\right)}^{2}}+\frac{1}{{\left(Bf_{t}^{\prime }\right)}^{2}\lambda_{o}}\lambda =C+A\lambda$$

After determining the empirical parameters, $Bf_{t}^{\prime }$ and $\lambda_{o}$, a double log-scale plot of the strength by structural size was generated, as shown in Figure 6. In the case of the Cauchy stress, the gradient was slightly less than 1:2 based on linear elastic fracture mechanics (LEFM) for the 10–30-mm specimens. However, an abruptly descending branch was observed in the case of the 40-mm specimen group. Thus, a significant drop in strength occurred in the case of the 40-mm specimens; this can be seen in Figure 5c,d. The reduction was extremely large in the context of linear fracture mechanics, where the gradient decreases proportionally to the square root.

Based on the work of Bažant [5,9], a linear softening branch was used to formulate the G_F_. Herein, a bilinear softening branch was used by modifying the area underneath the stress-strain curve in Figure 7. Thus, G_F_ can be rewritten as follows:

The equations of Bažant [4,5] can be rewritten as Equations (2) and (3), and are used to obtain the elastic stored energy in the FPZ shown in Figure 7b:(2)$$\frac{\partial W}{\partial a}=2(2k_{1}a+nd_{a})b\left(\frac{\sigma_{N}^{2}}{2E_{c}}\right),\;\frac{\partial W}{\partial a}=G_{F}b$$
(3)$$G_{F}=w_{c}\left(1-(1-\alpha^{2})\frac{E_{c}}{E_{t1}}-\alpha^{2}\frac{E_{c}}{E_{t2}}\right)\frac{f_{t}^{\prime 2}}{2E_{c}}$$
where, $k_{1}$ and n are the empirical constants derived in the experiments. According to Bažant [4,9], $k_{1}$ is close to 1 and n is 3~5, depending on the material used (3 for concrete; 5 for rocks and ceramics). a and $d_{a}$ are the notch length and maximum grain size, respectively. In our case, a varied among the specimens listed in Table 1. $d_{a}$ was 0.5 mm in the case of the sand particles used in the experiment. b is the thickness of the specimen, as listed in Table 1. W is the work done by the FPZ, which is shown in Figure 7b. $\sigma_{N}$ is the estimated tensile stress under the size effect and $f_{t}^{\prime }$ is the reference tensile strength. $E_{c}$, $E_{t1}$, and $E_{t2}$ are the initial elastic modulus and first and second descending softening branches, respectively, and α is the strength retention factor at the inflection point in the descending branch. $w_{c}$ is the crack band, which can also be expressed as $nd_{a}$. The inflection point, $\alpha f_{t}^{\prime }$ in Figure 7a ranges from 0.22–0.6$f_{t}^{\prime }$ depending on the specimen size; the average experimental value is 0.4$f_{t}^{\prime }$ (Figure 8).

The basic expression of B in Equation (1) can be rewritten as B* using Equations (4) and (5):(4)$$B^{*}=\sqrt{1-(1-\alpha^{2})\frac{E_{c}}{E_{t1}}-\alpha^{2}\frac{E_{c}}{E_{t2}}},\;f_{t}^{*}=\frac{f_{t}^{\prime }}{\sqrt[{}]{1+\frac{\lambda }{\lambda_{o}}}},\;\sigma_{N}=B^{*}f_{t}^{*2}$$
(5)$${\left(\frac{f_{t}^{\prime }}{\sigma_{N}}\right)}^{2}=\frac{1}{B^{*2}}+\frac{\lambda }{B^{*2}\lambda_{o}(\lambda )},\;Y=C+A\left(\lambda \right)\lambda$$
where, $b=\frac{1}{B^{*2}}$, $a(\lambda )=\frac{1}{B^{*2}\lambda_{O}\left(\lambda \right)}$, $\lambda_{O}\left(\lambda \right)=91.354-1.9073\lambda +0.0109\lambda^{2}$, $B^{*}$ = 3.29.

As mentioned earlier, $\lambda_{o}$ is not constant. Instead, we used the function λ to fit the experimental test results more accurately. Figure 9 and Table 2 show the inverse analysis results obtained using the size effect law defined in Equations (4) and (5), when the reference tensile strength was set to that of specimen T-40. The results are acceptable, being that they are between those given by the size effect law and the experimental results. However, they violated the linearized relationship of Equation (1) and a constant value of $\lambda_{o}$.

Bažant [4,5] described the size effect in terms of the member size, maximum aggregate size, FPZ, and crack width, w_c_, by first defining k_1_ and n. The size reduction rate depends on the 1:2 slope of the size radical, as given by Equation (1). As this law was intended for use in models of concrete and rock, it must be modified empirically for application to mortar. As shown in Figure 10, failure strength decreased with a gradient of ½ when n = 3, k_1_ = 1 and m = 2. However, there are discrepancies between this strength reduction and that shown by the experimental test data for the mortar specimen, where the slope value was 1/0.727. As mortar contains sand grains with a maximum size of 0.5 mm, w_c_ would be 1.5 mm if n = 3, as proposed by Bažant and Oh [8]. Then, the constants m and k_1_ have to be changed from 2 and 1 to 0.7 and 0.273, respectively, in accordance with the measured fracture energy (mode I fracture), as illustrated in Figure 7a. Here, $\lambda_{o}$ can be defined as $\lambda_{o}=\left(n/2k_{1}\right)\left(d/a\right)$.
(6)$$\sigma_{N}=B\cdot f_{t}^{*}$$
where, $f_{t}^{*}=\frac{f_{t}^{\prime }}{\sqrt[{m}]{1+\frac{\lambda }{\lambda_{o}}}}$, $\lambda_{o}=\left(\frac{n}{2k_{1}}\right)\left(\frac{d}{a}\right)$, $B=\sqrt{1+\frac{E_{c}}{-E_{t}}}$, $w_{c}=nd_{a}$.

As *n* = 3 and d/a = 4 is constant in $\lambda_{o}$, the acceptable range of tensile strength can be calculated with $\lambda_{o}$ = 16; then, k_1_ is close to 0.375 rather than ~1, as in the case of concrete [4,5]. This provides a better estimate of tensile strength with respect to specimen size, as shown in Table 3. If k_1_ is held constant at 1, then *n* should be changed to 8. However, this value far exceeds the range of 3~5 for concrete and rock. The lower value of k_1_ indicates that the FPZ in the vicinity of the notched crack would be narrower, as illustrated in Figure 7a. This could be explained by using a finite element smeared crack model with mode I fracture. In Figure 11b, the FPZ has a finite width (in red color) before developing into a major crack in the direction of the shortest length, as shown in Figure 11c. Regarding the minimum mesh size, h_ef_ (>w_c_), Bažant and Oh [8] assumed that the stress-strain relationship in the FPZ is linear, as follows:(7)$$w_{c}=\frac{2G_{F}}{f_{t}^{\prime }{}^{2}}{\left(\frac{1}{E_{c}}-\frac{1}{E_{t}}\right)}^{-1}$$
(8)$$w_{c}=\frac{2G_{F}}{f_{t}^{\prime }{}^{2}}{\left(\frac{1}{E_{c}}-(1-\alpha^{2})\frac{E_{c}}{E_{t1}}-\alpha^{2}\frac{E_{c}}{E_{t2}}\right)}^{-1}$$

## 5. Measurement of Mode I Fracture Energy Based on the Cauchy Stress

We developed a crack model based on the crack-opening law and fracture energy. This is suitable for modeling the propagation of cracks through concrete and was used in conjunction with the crack band. The crack opening function was derived experimentally by Hordijk [33], as shown in Equation (9).
(9)$$w_{f}=5.14\frac{G_{F}}{f_{t}^{\prime }}\;\mathrm{or}\;w_{f}≃2.06\frac{G_{f}}{f_{t}^{\prime }}\;\mathrm{where},\;\frac{G_{F}}{G_{f}}≃2.5$$

In our case, as shown in Table 4, the G_f_ is not affected by specimen size. The G_F_/G_f_ ratio showed that the mortar specimens were less brittle than other types of materials. Typically, this ratio is approximately 2.5 in the case of concrete [3,5]. The value of α, which links w_f_ and G_F_, is approximately 5.14 for concrete, whereas for mortar it is 3.71, and the average G_F_/G_f_ ratio is 1.94, as shown in Equation (10) and Table 4. Figure 12 shows the post-failure behavior of mortar according to specimen size. The definition of w_f_ proposed in Equation (10) was validated based on a nonlinear finite element simulation with a smeared crack model generated in ATENA software [34], as shown in Figure 13. Good agreement regarding the force and CMOD was seen between the simulation and experiment.
(10)$$w_{f}=\alpha \frac{G_{F}}{f_{t}^{\prime }}=2.55∼4.51\frac{G_{F}}{f_{t}^{\prime }}≃3.71\frac{G_{F}}{f_{t}^{\prime }}\;\mathrm{where},\;\frac{G_{F}}{G_{f}}=1.65∼2.23≃1.94$$

Hillerborg [35] and Bažant [36] calculated l_ch_ based on the G_F_ and G_f_. As given by Equation (9), the ratio between G_F_ and G_f_ is approximately 1.94 in the case of our mortar specimen. These two characteristic lengths have the same change with the factor of 1.94 in Equation (9). These measures of l_ch_ represent the brittleness of the specimen in terms of its elastic energy, U_o_ with respect to the surface energy released by crack development, W_s_, as indicated by Equation (11). In Figure 14, we can see the decrease in brittleness with larger specimen size.
(11)$$l_{ch}=\frac{E_{c}G_{F}}{f_{t}^{\prime }{}^{2}}=1.94\frac{E_{c}G_{f}}{f_{t}^{\prime }{}^{2}}=\frac{W_{s}}{U_{o}}$$

## 6. Conclusions

In this study, we investigated the fracture energy and size effect of mortar specimens under direct uniaxial tension.The fractured surfaces were scanned using a 3D scanner and we calculated Cauchy stresses to evaluate the fracture energy precisely by using final fractured surface. The actual fractured surfaces were larger than the initial section, so the fracture stresses were calculated according to the Cauchy stress principle. It is important to quantify the mode I fracture energy based on experimental test data. Smaller specimen group shows relatively larger differences between fractured and initial sections.The average G_F_/G_f_ ratio for the mortar specimens was 1.94, which is lower than the typical value for concrete, of 2.5. The direct tension test on the double-notched mortar specimens with no major inclusion yielded a small FPZ after tensile crack initiation, so that the tail portion of the softening branch was also relatively small. This led to a decrease in the G_F_/G_f_ ratio. We verified this via a nonlinear fracture mechanics simulation, which agreed well with our experimental results.We investigated the size effect for four different specimens with d/a held constant, to eliminate shape effects. The traditional LEFM-based prediction and the Bažant size effect law predict a gradient of 1/2 in the case of relatively large specimens. Typical laboratory-scaled specimens usually exhibit gradients below 1/2 in the case of concrete; thus, nonlinear fracture mechanics should be used to estimate tensile strength in terms of structural dimensions. In the case of our mortar specimens, the slope value, 1/0.727, violated either the LEFM theory and Bažant size effect law, which was unexpected because LEFM predicts a strong size effect. More observations are required to explore this according to variations in d/a. In order to explore this discrepancy, different direct tensile test specimen could be applied; (a) un-notched specimen, (b) one-side single-notched specimen, and (c) double-notched specimen. Notch sensitivity may lead to different extent of FPZ and tensile strength change in accordance with specimen size.

## Figures and Tables

**Figure 1 materials-12-01850-f001:**
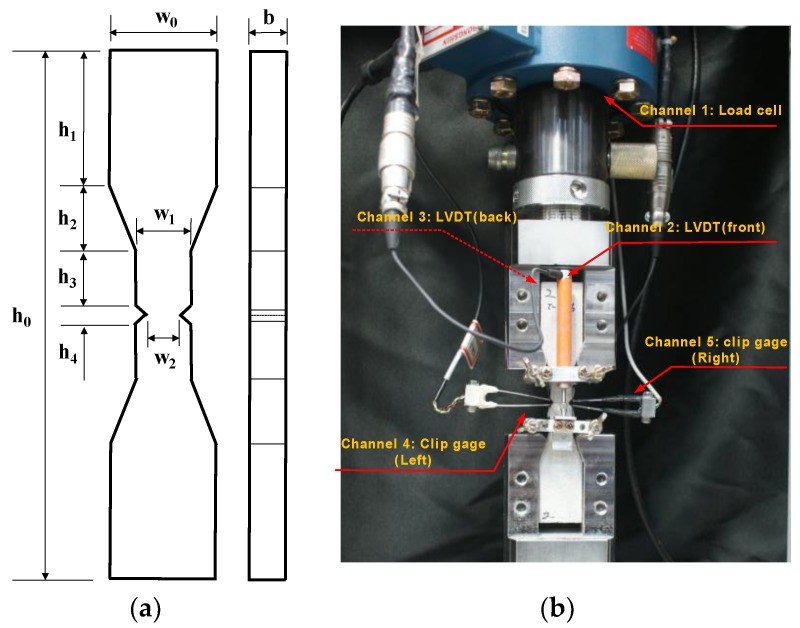
(**a**) Generic geometry of the specimens and (**b**) test apparatus.

**Figure 2 materials-12-01850-f002:**
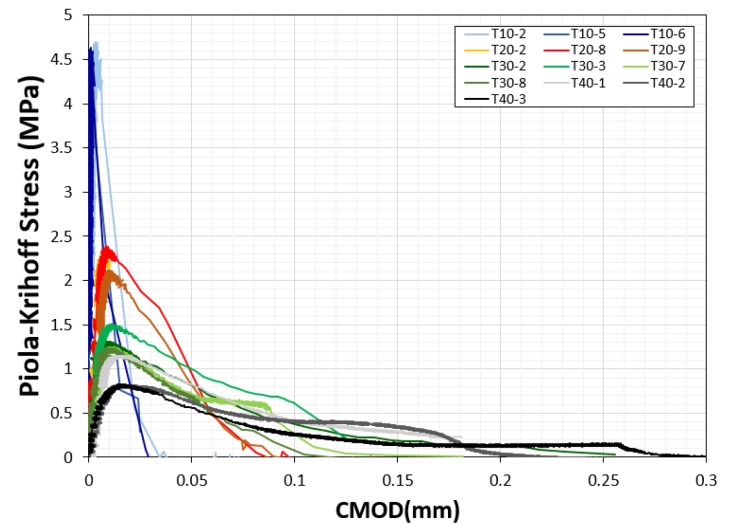
Plot of the Piola-Krichhoff (PK) stress versus the crack mouth opening displacement (CMOD).

**Figure 3 materials-12-01850-f003:**
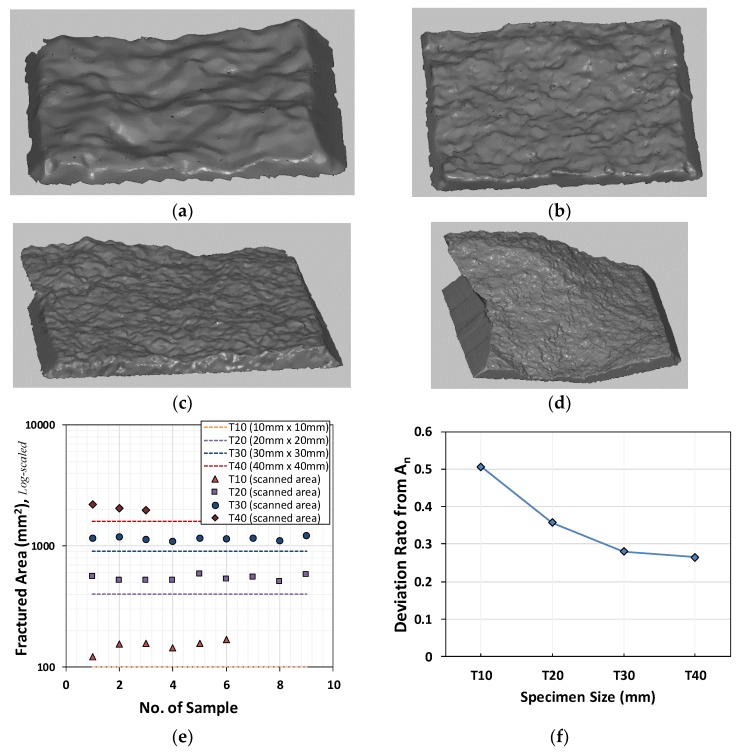
Scanned fractured sections. (**a**–**d**) Scanned images; (**a**). T-10 (A_n_ = 144.67 mm^2^); (**b**). T-20 (A_n_ = 554.05 mm^2^); (**c**). T-30 (A_n_ = 1158.09 mm^2^); (**d**). T-40 (A_n_ = 2,217.54 mm^2^); (**e**) measured fractured surface areas by specimen size; (**f**) deviation ratios between fractured and initial sections.

**Figure 4 materials-12-01850-f004:**
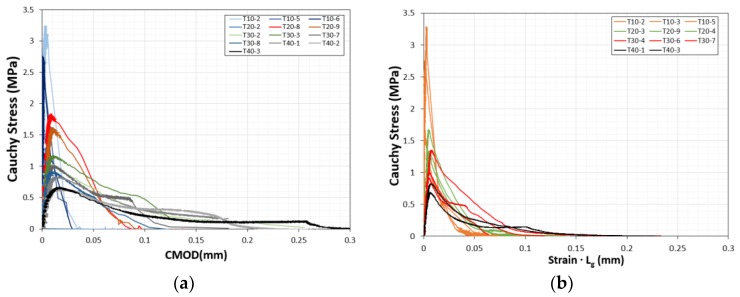
Direct tensile strength-CMOD relationship based on the (**a**) clip gages and (**b**) linear variable differential transformers (LVDTs).

**Figure 5 materials-12-01850-f005:**
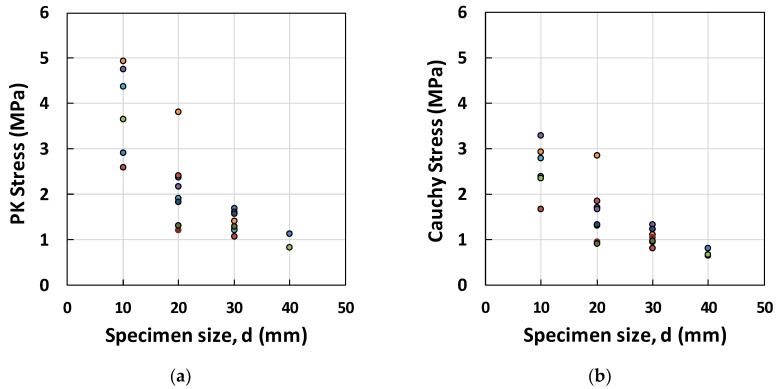
Peak stresses versus specimen size: (**a**) PK, and (**b**) Cauchy stresses, and linearized stresses versus different specimen sizes: (**c**) PK, and (**d**) Cauchy stresses.

**Figure 6 materials-12-01850-f006:**
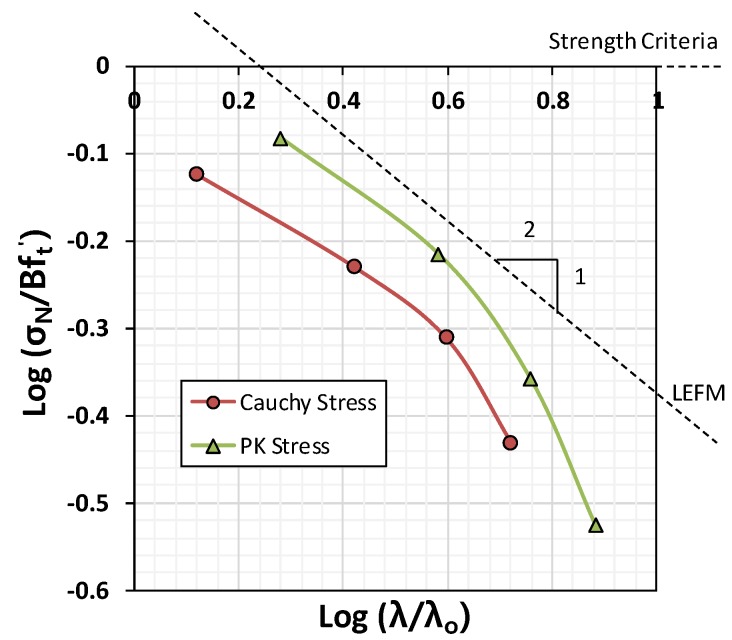
Double log-scale plot of the size effects.

**Figure 7 materials-12-01850-f007:**
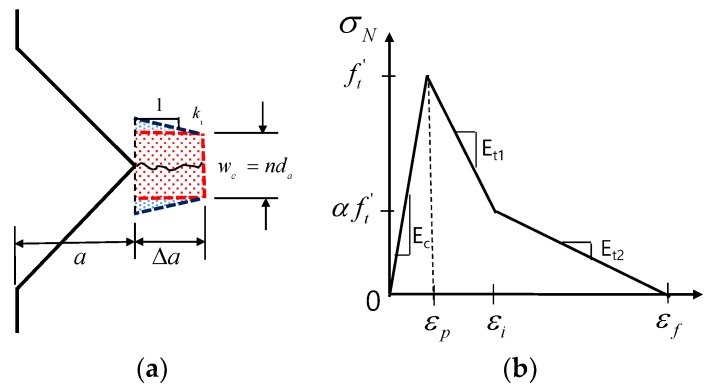
Fracture process. (**a**) Fracture process zone (FPZ) and (**b**) bilinear softening model for mortar.

**Figure 8 materials-12-01850-f008:**
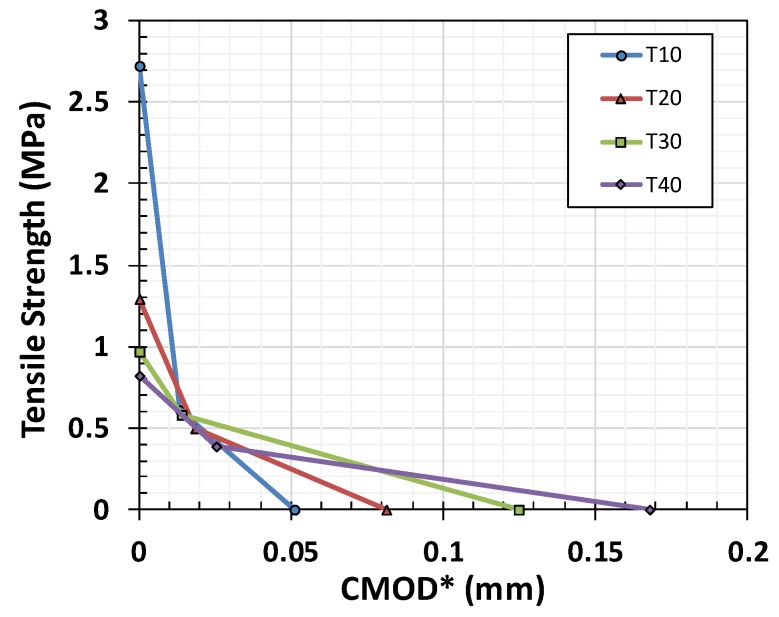
Simplified stress-strain relationship based on the average tensile stress-inelastic CMOD for measuring specific fracture energy (G_f_) and total fracture energy (G_F_).

**Figure 9 materials-12-01850-f009:**
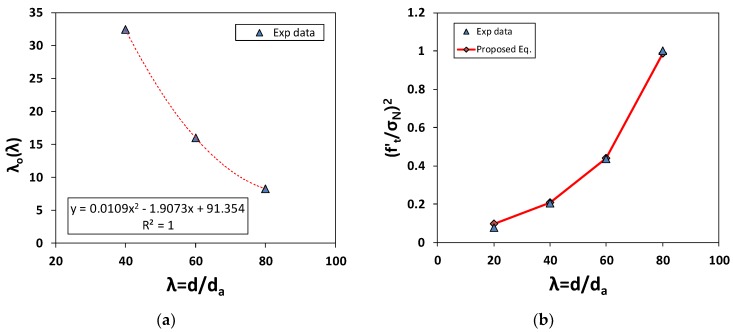
(**a**) The material characteristic length, λ_o_ with respect to the structural size, λ; (**b**) the size effect when λ_o_(λ).

**Figure 10 materials-12-01850-f010:**
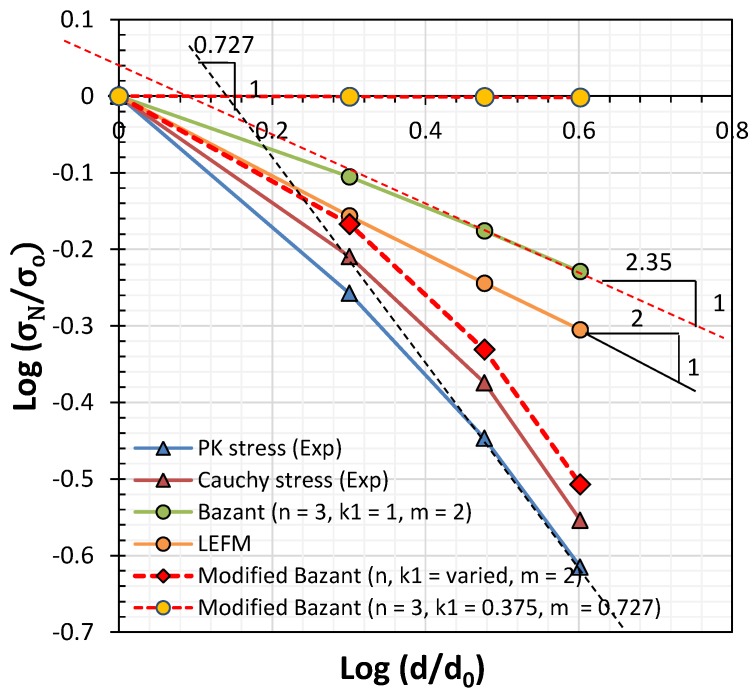
Doubly log-scale plot of the size effect of the T-series mortar specimens.

**Figure 11 materials-12-01850-f011:**
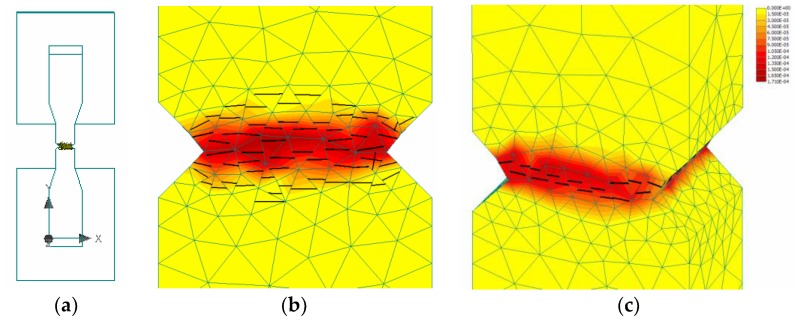
Llustration of the FPZ and failure simulation performed with ATENA software: (**a**) FE model of T-series specimen with the rigid fixtures at the ends, (**b**,**c**) crack distribution and FPZ across the notches with different crack width.

**Figure 12 materials-12-01850-f012:**
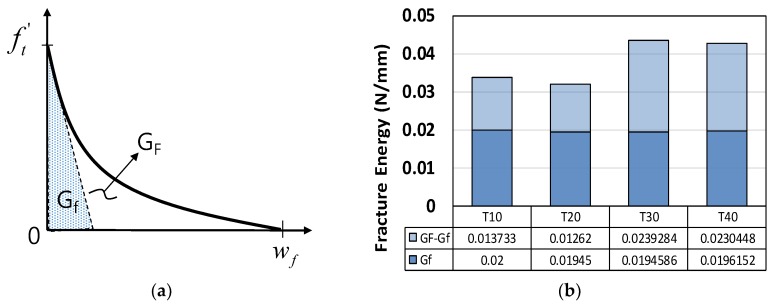
Fracture energies: (**a**) specific and total fracture energies, G_f_ and G_F_, (**b**) measured fracture energies from T-series specimens.

**Figure 13 materials-12-01850-f013:**
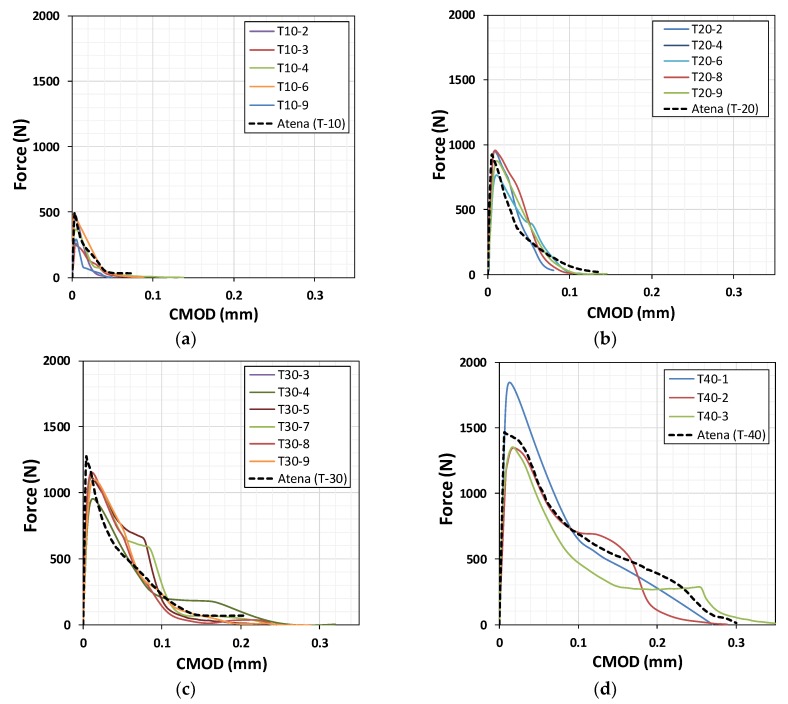
ATENA simulation results using the measured values of G_f_ and G_F_ for specimens (**a**) T-10, (**b**) T-20, (**c**) T-30, and (**d**) T-40.

**Figure 14 materials-12-01850-f014:**
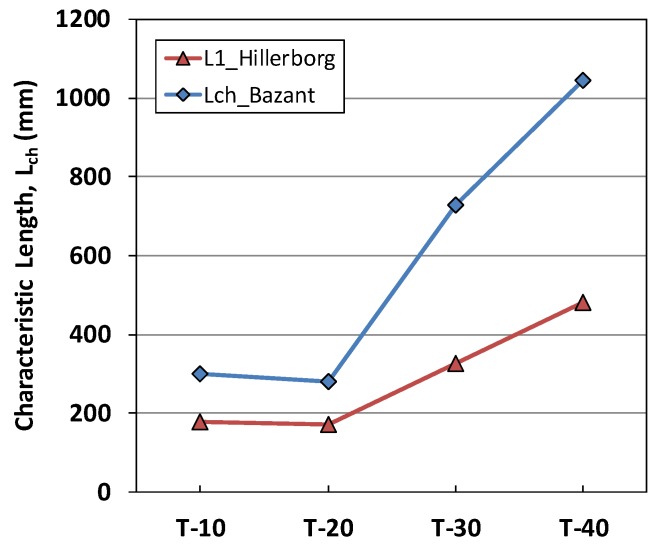
Characteristic length of the T-series specimens.

**Table 1 materials-12-01850-t001:** Specimen dimensions in mm (Note: structural dimension, d = w_2_).

Specimen Label	h_0_	h_1_	h_2_	h_3_	h_4_	w_0_	w_1_	w_2_	a	b	A_o_ (mm^2^)
T-10	165.0	42.5	20.0	17.5	5.0	30.0	15.0	10.0	2.5	10.0	100
T-20	330.0	85.0	40.0	35.0	10.0	60.0	30.0	20.0	5.0	20.0	400
T-30	495.0	127.5	60.0	52.5	15.0	90.0	45.0	30.0	7.5	30.0	900
T-40	660.0	170.0	80.0	70.0	20.0	120.0	60.0	40.0	10.0	40.0	1600

**Table 2 materials-12-01850-t002:** Strength variation by specimen.

Specimen Label	Avg. Cauchy Stress (MPa)	Estimated Tensile Stress (MPa)
T-10	2.577	2.327
T-20	1.591	1.583
T-30	1.088	1.085
T-40	0.719	0.723

**Table 3 materials-12-01850-t003:** Log-scale data of the size effect of the T-series mortar specimens.

Specimen Label	PK Stress	Cauchy Stress	LEFM	Bažant (*n* = 3, *k*_1_ = 1, m = 2)	Modified Bažant (*n* = 3, *k*_1_ = 0.375, m = 0.727)
T-10	3.89	2.58	3.16	4.09	2.78
T-20	2.15	1.59	2.20	3.21	1.51
T-30	1.39	1.09	1.80	2.73	0.99
T-40	0.94	0.72	1.56	2.41	0.72

**Table 4 materials-12-01850-t004:** Value of parameter α used for calculating crack bandwidth, w_f_.

Specimen Label	f_t_^’^(MPa)	w_f_ (mm)	G_f_ (N/mm)	G_F_ (N/mm)	G_F_/G_f_	α
T-10	2.58	0.0333	0.02	0.0337	1.685	2.55
T-20	1.59	0.0910	0.01945	0.0321	1.650	4.51
T-30	1.09	0.1450	0.01946	0.0434	2.230	3.64
T-40	0.72	0.2450	0.01962	0.0427	2.176	4.13

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
