# Peer review of "Fracture Parameters of Cement Mortar with Different Structural Dimensions Under the Direct Tension Test"

_materials, 2019, doi:10.3390/ma12111850_

Reviewer 1 Report

Given the extensive amount of published research on the size effects of concrete and related materials like mortar, there should be a clear case for reexamining the subject. Unfortunately, the paper does not present a clear case. Improving readability of the introduction and other key parts of the paper should help. As noted below some of the conclusions report on expected behaviors.

The introduction is one long "paragraph", which makes it impenetrable. Proper use of paragraph structure would improve readability. At the end of the introduction, the objectives of the research are not clear. Since size of effects of mortar and concrete specimens have been examined extensively in the literature over the past 40 years, the novelty and relevancy of the research plan should stand out more clearly.

Conclusion 1: what is the relationship between laser scanning of the fracture surfaces and calculating the Cauchy stresses? The logic is not clear.

Conclusion 2: the observed behavior for mortar specimens, relative to that expected for concrete specimens, is expected. What is the significance of this conclusion, which describes an expected outcome?
Conclusion 3: the 1/2 slope asymptote for large structures is explained by theory and most large scale experimental results show that trend. The authors' slope of 1/0.727 is difficult to explain. The suggestions given in the paper are not supported with any technical clarity.

Author Response

The authors appreciate the valuable comments on the manuscripts. Please find the attachment to see the author's response.

Reviewer 2 Report

The authors present a research work related to “Fracture Parameters of Cement Mortar with Different Structural Dimensions under Direct Tension Test” where the fracture energy and size effect of mortar specimens under direct uniaxial tension was investigated.

Remarks to the authors:

1. Please relate how your results can be interpreted in the context of previous recent studies.

2. Future research directions may also be mentioned.

3. References should be described according to the manuscript instructions.

Author Response

The authors appreciate the valuable comments on the manuscripts. Please find the attachment to see the author's response.

Round  2

Reviewer 1 Report

The revised version of the paper has not addressed some of the review comments. In particular:

The introduction is still one long paragraph. The authors were instructed to break the introduction into several readable paragraphs, but that was not done. Please consult textbooks on technical writing about making effective paragraphs.

The work should be introduced with detailed descriptions of how it relates to past publications in this highly covered subject of size effect of concrete structures. The authors added citations to the paper, but did not make reasonably good connections between the cited material and their research. It is not effective to simply add references and cite them in bulk form, as done in the introduction by [26-42].

References that do not clearly support understanding and objectives of the paper should be removed. The bulk form citation [26-42] is counterproductive to understanding the paper.

The authors maintain that size effect has not been investigated, or there has been few investigations, for the case of uniaxial tension. However, there have been many instances of size effect testing using tension specimens. For one, the authors should consult MRA van Vliet, Size effect in tensile fracture of concrete and rock, PhD thesis, Delft University Press, Delft, The Netherlands, 2000. This research also appears in journal papers and in the textbook of JGM van Mier, CRC Press.

The authors try to address the review question about the 1/0.7 slope by saying additional tests should be run. However, they kept the statement "One possible explanation for this result...", which is not supported by rational analysis.

The paper will probably need to undergo additional review(s) after this set of comments are addressed.

Author Response

The authors appreciate the reviewer's valuable comments. Please find the attachment.
